# Towards a Human-Centric Digital Twin for Human–Machine Collaboration: A Review on Enabling Technologies and Methods

**DOI:** 10.3390/s24072232

**Published:** 2024-03-30

**Authors:** Maros Krupas, Erik Kajati, Chao Liu, Iveta Zolotova

**Affiliations:** 1Department of Cybernetics and Artificial Intelligence, Faculty of EE & Informatics, Technical University of Kosice, 042 00 Kosice, Slovakia; erik.kajati@tuke.sk (E.K.); iveta.zolotova@tuke.sk (I.Z.); 2College of Engineering and Physical Sciences, Aston University, Birmingham B47ET, UK

**Keywords:** human–machine collaboration, digital twin, human-centric, enabling technologies and methods, Industry 5.0

## Abstract

With the intent to further increase production efficiency while making human the centre of the processes, human-centric manufacturing focuses on concepts such as digital twins and human–machine collaboration. This paper presents enabling technologies and methods to facilitate the creation of human-centric applications powered by digital twins, also from the perspective of Industry 5.0. It analyses and reviews the state of relevant information resources about digital twins for human–machine applications with an emphasis on the human perspective, but also on their collaborated relationship and the possibilities of their applications. Finally, it presents the results of the review and expected future works of research in this area.

## 1. Introduction

During the advancements of industrial revolutions, the concept of the digital twin (DT) has emerged, revolutionising industries and redefining our approach to design, monitoring, and maintenance. The DT is a concept that has many definitions [1], but essentially, it is a virtual digital replica or simulation of a physical entity, be it a product, process, or system. This digital counterpart is created by integrating real-time data from sensors, Internet of Things (IoT) devices, and other sources, providing a dynamic and accurate representation of the physical entity’s behaviour, while also being a powerful tool for analysis, optimisation, and decision making throughout the entire lifecycle of the corresponding physical entity.

The new trend towards human-centric manufacturing aims to place humans at the centre of manufacturing systems and processes [2]. The concept of human–machine collaboration (HMC) emerged before Industry 5.0 as a solution to combine the strengths of both machines and humans, having great potential for fulfilling human needs and easing physically and mentally demanding tasks. However, most HMC applications created are system-centric, focusing on effectiveness and productivity, rather than on humans. Because of this, HMC should also evolve to be more human-centric [3]. Industry 5.0 emphasises the importance of human workers alongside advanced technologies. HMC in industry often occurs in complex workspaces that need to adapt regularly.

The DT, together with other enabling technologies, is a solution to manage these complex systems by creating digital counterparts of these workspaces and by altering them based on our needs. By creating a DT of the HMC workspace, we can manage individual tasks and problems more intuitively. However, similar to HMC, DTs have primarily focused on productivity rather than on human-centric aspects, presenting a challenge in creating easily accessible human-centric applications. The value-driven approach of Industry 5.0 requires us to shift our focus and create a systematic approach to creating human-centred solutions.

In the transition to Industry 5.0, human-centric DTs are pivotal in creating explicit connections between humans and technologies to complement their strengths in HMC applications. This research aims to develop a systematic approach to creating human-centric, DT-driven HMC solutions. By bridging the gap between DTs and HMC, we can unlock the potential for symbiotic human–machine applications that prioritise both productivity and the quality of work life for human operators. Consequently, there is a pressing need to define how to utilise DTs to make more human-centric solutions for HMC. To address this gap, we investigate the current state of research on human-centric applications involving the use of DTs in HMC. Additionally, we identify enabling technologies and methods that facilitate human-centric applications involving DTs and HMC.

## 2. Background

Digital twins play a pivotal role in facilitating human–machine collaboration in Industry 5.0. They offer intuitive interfaces, real-time insights, and collaborative capabilities, empowering human operators to optimise processes and make informed decisions. This symbiotic relationship underpins Industry 5.0’s transformative potential, enabling organisations to excel in a digital world.

This section provides background literature for the key concepts of this paper. It begins with an exploration of Industry 5.0, highlighting its human-centric approach and the need for a systematic approach to manage enabling technologies (Section 2.1). The section then delves into HMC, emphasising various types of relationships and the challenges faced in designing human-centred workspaces (Section 2.2). Lastly, the concept of the DT is introduced, showcasing its role in facilitating interactions between humans and machines and the evolution towards human-centric DTs (Section 2.3).

### 2.1. Industry 5.0

Since the introduction of the concept of Industry 5.0, researchers have tried to define and agree on how to realise its values, including human-centric applications in various areas of industry. The literature is mostly focused on identifying human needs and what technologies could be used together to fulfil those needs. However, these solutions are often partial, lacking a systematic approach for integrating these technologies to create comprehensive human-centric applications.

Firstly, the Industry 5.0 document by the European Commission [4] presented six categories of enabling technologies, consisting of subcategories, and stated that the full potential of the mentioned technologies can be achieved by using the presented technologies together in a synergistic manner. Regarding the topic of this review paper, the document mentions “individualised human–machine-interaction” and “digital twins and simulation”; however, it does not state how to use the mentioned technologies to accomplish its values, leaving space for researchers’ interpretations [3,5,6,7]. Since Industry 5.0 complements the previous Industry 4.0 and their enabling technologies cross paths [8], it is clear that many enabling technologies of Industry 4.0 will also undoubtedly help to achieve the societal goals of Industry 5.0 [9].

The human-centric approach in industry puts human needs and interests at the centre of the processes. It also means ensuring that new technologies do not interfere with workers’ fundamental rights, such as the right to privacy, autonomy, and human dignity. Humans also should not be replaced by robots in industry, and they should synergistically combine with machines to improve workers’ health and safety conditions [10]. Starting with the previous fourth industrial revolution, human-centric solutions and concepts were also created, for example Operator 4.0 [11], focusing on human–automation symbiosis, which has now started its transition towards Operator 5.0 under Industry 5.0’s influence [12]. Similarly, the concept of human–cyber–physical systems (HCPSs) [13], a composite intelligent system comprising humans as operators, agents, or users, is also evolving towards human-centric manufacturing. Regarding human-centricity in Industry 5.0, the authors in [3] presented an industrial human needs pyramid for Industry 5.0, focusing on higher human needs like belonging, esteem, and self-actualisation. They state that human-centric manufacturing should go beyond traditional human factors and focus on a higher humanistic level, such as cognitive and psychological well-being, work–life balance, and personal growth.

### 2.2. Human–Machine Collaboration

HMC is one type of relationship between a human and a machine. Both the term and this type of relationship emphasise both humans and machines collaborating on the same tasks and goals simultaneously, allowing robots to leverage their strength, repeatability, and accuracy, while humans contribute their high-level cognition, flexibility, and adaptability. It involves parties with different capabilities, competencies, and resources, which must be coordinated to maximise their strengths. Other most defined human–machine relationships are coexistence and cooperation [3,14]. Apart from collaboration, human–machine relationships like cooperation, where humans and machines can temporarily share their resources or workspace, they also may work on the same goal, but have their own tasks. The first-ever relationship, coexistence, is where humans and machines do not share their workspace at all. Possible future relationships have also been defined, like coevolution and compassion [3].

In this paper, we give the word “machine” a broad meaning as it can refer to an automated or autonomous system, an agent, a robot, an algorithm, or an AI. Therefore, studies on HMC span a variety of fields, including human–robot collaboration (HRC), human–machine interaction (HMI), or human–machine teaming, involving extensive literature on robotics. While HMC is focused on synergy and combined effort, HMI refers to any situation when humans interact with machines and does not necessarily involve collaboration or working on a common goal [15]. One of the major goals of the field of HMI is also to find the “natural” means by which humans can interact and communicate with machines [16].

Many studies on HRC have already focused on effective collaboration between humans and cobots, which led to the creation of collaborative assemblies [14,17]. Cobots or collaborative robots are designed to interact physically with humans in a shared environment without barriers or protective cages. These applications also proved useful to be extended and used with their DT, which enabled real-time control and dynamic skill-based task allocation. Using predictions and simulations led to optimised behaviour without the risk of human injury or financial loss. However, these applications and studies were limited only to cobots, which are usually implemented in closed industrial cells.

HMC today faces many identical and social challenges addressed in the literature [3,5,18,19], such as transparency, explainability, technology acceptance and trust, safety, performance measures, training people, or decision-making risks while using AI [20]. Overcoming these challenges would result in improved human well-being and flexible manufacturing, where humans and machines develop their capabilities. In [19], the authors proposed a framework with guidelines and recommendations for three complexity levels of the influencing factors presented when designing human-centred HMI workspaces in an industrial setting. It was concluded that challenges from designing HMC workspaces in industrial settings require multi-disciplinary and diverse knowledge of fields with a framework to systematise research findings.

Working alongside cobots appears to be an effective method for creating personalised products, yet it also raises important issues and considerations that need to be tackled. As mentioned in [5], these concerns encompass fears of job loss among humans, psychological issues, and the challenge of dynamic task distribution. The authors noted that HRC in factories is more successful when cobots assist in repetitive tasks, allowing humans to focus on creative and innovative work.

### 2.3. Digital Twin

The concept of the DT was proposed in 2002, but became a reality due to the surge of IoT devices, which are used for collecting vast amounts of data, thus making DT accessible and affordable for many companies. Based on the collected data, it is possible to analyse and monitor the digital counterparts in real-time to make decisions or prevent problems in the physical world [5], making DTs essential to improve interactions between humans and machines [21]. Thus, to enable the efficient design, development, and operation of an HRC system, some DT frameworks were already created [22]. In the recent literature [23], the authors identified six key application areas for DTs with strong human involvement, one of which was ergonomics and safety, as well as other identified areas such as training and testing of robotics systems, user training and education, product and process design, validation, and testing.

During the past few years, there has been a trend to combine DTs with semantic technologies to enhance them with cognitive abilities [24]. From this trend, the concept of the Cognitive Digital Twin (CDT) [25,26] emerged as a promising next evolution stage of DTs, which can replicate human cognitive processes and execute conscious actions autonomously, with minimal or no human intervention. Besides cognitive abilities, the CDT should have multiple levels and lifecycle phases of the system. Key enabling technologies for the CDT are semantic technologies (ontology engineering, knowledge graphs), model-based system engineering, product lifecycle management, and industrial data management technologies (cloud/fog/edge computing, natural language processing, distributed ledger technology) [24]. For HRC cases in smart manufacturing, ref. [27] proposed a CDT framework and case study to learn human model knowledge through deep learning algorithms in edge–cloud 5G computing to improve the interaction, facilitating workers’ lives. Similarly, the concept of a Digital Triplet was created [28] containing the cyber world, physical world, and “intelligent activity world”, where humans solve various problems by using the DT.

Based on the reviewed literature, one of the main parts of creating human-centric DT sin HMC should involve creating a DT of a human [3,29]. Currently, most existing DT applications are developed for prediction and monitoring purposes to be used as decision-making applications for humans, and the importance of human involvement in the DT environment is overlooked, as past research is mostly focused on manufacturing devices, which creates one of the main research issues [29]. The authors in [3] also mention the importance of creating a human digital twin (HDT), which can be created according to a worker’s capabilities, behaviour pattern, and wellness index. One technical limitation in creating an HDT arises from the varied methods of communication. A DT of an electronic object can exploit real-time communication with its physical counterpart, but a DT of a human connects with its physical twin through intermediary devices, typically sensors or software applications [30].

## 3. Review Methodology

Based on the title search in the Web of Science (WoS) database for review categorisation, the numbers of review and survey papers on themes such as Industry 5.0, HMC or HRC, or DT, we can see that the trend of these concepts has increased rapidly in the last 5 years Figure 1. However, in the case of DT, most of the review and survey papers are scarcely or not at all focused on human factors. For this review, to evaluate previous review papers, we decided to select and focus only on those articles that were focused on humans in more depth, which resulted in a comparison of 20 review and survey papers that were found in the WoS database for the keyword- and title-based search as follows:


*(TI = (digital twin*) OR AK = (digital twin*)) AND (TI = (human* OR industry 5.0 OR operator* OR user* OR people OR worker* OR employee*) OR AK = (human* OR industry 5.0 OR operator* OR user* OR people OR worker* OR employee*))*


The search phrase was altered to contain the most common phrases labelling humans in various areas of industry. This also involves papers with topics such as HMI, which is also a part of the HMC scenarios. The search phrase found 29 articles related to the topic of our review paper, from which we picked only 20 more closely related articles, see Table 1. Each article was then rated based on its primary focus on coverage, as low, medium, or high, to evaluate how well it focused on enabling technologies, certain methods for these technologies, their use cases, and to what extent they also cover human–machine system topics. The authors of the papers propose challenges and future perspectives of DTs for futuristic human-centric industry transformation.

In the next phase, we picked all articles regarding our search phrase, not restricted only to reviews and surveys, and based on 422 results, we created a keyword co-occurrence map in VOSViewer to further help identify enabling technologies and methods for this topic; see Figure 2. Keywords from the network map are shown and ordered by occurrences in Table 2.

In Table 1, we can see that more than half of the related review papers chosen for comparison were published from the year 2023. After reading the papers, we concluded that some papers had satisfying higher coverage on topics related to this review paper. Still, none of them focused directly on the enabling technologies and methods of DTs for HMC regarding human-centric topics and, therefore, lacked proper depth, as they tended to focus on specific technologies or different topics.

The found keywords in Table 2 and their total link strength show that HRC is the biggest topic related to humans and DT, while the term “human–machine collaboration” is not used extensively in the literature. The technologies with the biggest total link strength include virtual reality and artificial intelligence, indicating their biggest literature coverage in implementing DTs in HMC applications.

Our background literature review and review methodology underscore a notable research gap in the current literature, indicating a pressing need for more comprehensive studies addressing the human-centric aspects of DT-driven HMC solutions. The question of how DT integration can enhance the development of human-centred HMC applications, along with identifying the most suitable technologies and methods for optimising this integration, remains unanswered. Therefore, key enabling technologies, their methods, and paradigms for DT for HMC are discussed and analysed in the next sections as existing applications, concepts, and use cases are explored and analysed from the human-centric point of view.

## 4. Enabling Technologies and Methods

This section discusses the importance of various enabling technologies presented in an official document of Industry 5.0 [4] and the discovered keywords from Section 3 while focusing on the use cases of these technologies. This includes digital twins and simulations (Section 4.1), artificial intelligence (Section 4.2), human–machine interaction (Section 4.3), and data transmission, storage, and analysis technologies (Section 4.4), often used together in many applications. This section will concentrate on integrating DTs with these technologies in HMC scenarios and address applications with similar topics.

### 4.1. Digital Twins and Simulations

Before creating a DT, it is necessary to decide on the appropriate tools. According to [49], the five-dimensional DT model can provide reference model support for applications of DT in different fields. Based on the five dimensions, the authors created a reference framework of enabling technologies for DTs, from which we include technologies for cognising and controlling the physical world, DT modelling, data management, services, and connections. Different tools support different sets of features and technologies; therefore, it is up to engineers to choose a tool that will fulfil the needs of our application. Based on our review methodology, we searched articles focusing on DT implementations and use cases with various technologies.

In Table 2, we focus on describing the most common tools we found in the literature to create DT-driven HMC applications. These tools vary in their application areas and, therefore, were used for different application problems. The cost of the software needs to also be considered when selecting commercial tools, including the cost associated with training to be able to learn to use such tools. Some tools, such as the Robot Operating System (ROS), are used, which is an open-source middleware, thereby offering a cost-effective communication framework for DT applications. ROS-compatible software, such as CoppeliaSim, Gazebo, RViz, Unity, and Blender, is often implemented as a simulation platform, although some of them are rather game engines than a simulation environment [44]. Unity, the most used tool in our review, is a proprietary software for which buying a personal license is not mandatory, similar to V-REP, where the education edition is for free. However, tools such as Technomatix Process Simulate or Matlab can be costly. In some cases, custom tools for simulation were also created.

The most common implementation areas for HMC applications included safety and ergonomics, maintenance, task planning, optimisation, testing, and training. Some articles, while having humans in the loop, were focused more on increasing productivity and optimisation. A variety of methods are utilised in the literature for the implementation of DTs to facilitate safe interactions of robots with human operators and optimise ergonomics during such interactions. Decision making supports task planning and allocation and gives operators autonomy. Human action recognition and prediction are also used to optimise production or ensure human safety. With the help of the simulation technologies, the authors also focused on designing and testing safe HMC workspaces before their implementation, which simplifies training and robot programming.

Choosing the right DT tools, as identified in Table 3, is crucial for implementing other key enabling technologies for HMC, such as AI, and HMI technologies like extended reality (XR). However, a great variety of methods and techniques for these technologies can make choosing the right ones for certain use cases complicated. Identifying the most and least common use cases for DT technologies in the HMC domain can help organisations determine how they can best leverage the technology to improve their operations and processes and start to put humans in the centre. Therefore, the next subsections will discuss enabling technologies and methods in various use cases and analyse how they enable human-centric DT in HMC applications.

### 4.2. Artificial Intelligence

AI is one of the enabling leaders and facilitators for growth and adaptability for DTs, also becoming the main component of such systems [68]. Based on the literature review, we identified specific methods and tasks for different application problems, as seen in Table 4. In the literature focusing on humans, authors focused on topics such as safety, ergonomics, decision making, and training. Still, simultaneously, some authors focused also on increasing the effectiveness of HMC applications in areas where the focus on humans is not as big, such as robot learning, assembly line reconfiguration, or optimisation.

Most applications involving human safety use deep learning for object detection or recognition for collision avoidance or calculating the distance between humans and robots. Many DT applications use AI as a tool to optimise processes focusing on task planning or decision making. Some authors started to focus on recognising human behaviour, especially by motion detection, to be able to predict the next human action, which has wide application potential. Artificial Neural Networks are often used for computer vision-related tasks, which often involve the use of HMI devices such as Red Green Blue-Depth (RGB-D) cameras to gather data to train neural networks for tasks such as object recognition or human motion detection, mainly for solving human safety and ergonomics. Computer vision can also help with creating HDTs, as applications may involve the recognition of human facial features, expressions, poses, and gestures [23].

In [60], the authors developed a solution to monitor operator’s ergonomics by implementing a Convolutional Neural Network (Single-Shot Detector) to detect the existence or position of parts to be assembled, as well as the hands of the operator. The neural network model was trained on real-life and virtual object photos and increased its accuracy by also training on more synthetic data. In the next work, decision making based on artificial intelligence logic is used to derive alternative production system configurations, such as the optimal layout and task plans to reconfigure the system in cases of unexpected events online [59].

In [69], the authors proposed using Visual Question Answering (VQA) in HMC applications to increase effectivity and safety. VQA is a method for video understanding, consisting of computer vision and natural language processing algorithms, making it a multimodal method, consisting of more artificial intelligence methods. In some applications, the composition of multiple modules of AI is used, resolving in composable or composite AI [70]. In [54], the authors fused different human-tracking sensors and combined deep learning with semantic technologies for predicting human interactions in HRC.

AI algorithms can process and analyse large volumes of data collected from sensors, machinery, and various other sources. By extracting crucial insights and identifying patterns within these data, these algorithms can assist manufacturers in making more informed decisions and enhancing the efficiency of their operations. One of the advantages of AI-equipped DTs over conventional ones is the ability to better respond to HMC applications due to the uncertainties in the environment and sensors and the randomness and diversity of human behaviour [71]. Implementing AI in HMC can advance learning processes, allowing humans and machines to adapt to changing environments or requirements.

**Table 4 sensors-24-02232-t004:** Artificial intelligence methods used in DT HMC applications.

AI Method	Specific Technique	Task	Application Problem
Traditional methods	SVM [50,72]	Classification [50] Object recognition [72]	Human skill level analysis [50] Soft-robot tactile sensor feedback [72]
Heuristic methods	Search algorithms [59]	Decision making [59]	Assembly line reconfiguration and planning [59]
Neural networks	FFN [57] RNN [73]	Object recognition [57] Sequential data handling [73]	Human safety [57] Dynamic changes’prediction [73]
Deep learning	1D-CNN [74] Mask R-CNN [75,76] CNN [53,58,60,77,78] PVNet Parallel network [79] PointNet [79] SAE [80]	Detection or recognition [53,60,74,75,76,79] Classification [77] Human action and motion recognition [78] Pose estimation [58,79] Feature extraction [73] Anomaly detection [80]	Human safety [60,75,76,78,79,80] Ergonomics [60] Human action andintention understanding [60,77] Efficiency [78,79] Position estimation [53] Decision making [73] Object manipulation [74]
Reinforcement learning	Model-free RL[51] TRPO [61] PPO [61] DDPG [61] Q-learning [63]	Robot motion planning [51] Robot learning [61] Dynamic programming [63]	Training [51] Teleoperation [51] Robot skill learning [61] Assembly planning optimisation [63]
Deep reinforcement learning	Deep Q-learning [81] PPO [52] SAC [52] DDPG [64] D-DDPG [73]	Task scheduling [81] Decision making [81] Training [52] Humanoid robot arm control and motion planning Optimisation [64,73]	Smart manufacturing [81] Optimisation [52] Robot learning [64] Enhancing efficiency and adaptability [73]
Generative AI	motion GAN [82]	Human motion prediction [82]	Human action prediction [82]

### 4.3. Human–Machine Interaction

Since humans interact with DTs in both physical and virtual worlds, human–computer interaction technologies and human–machine interaction should be incorporated. Similar to other technologies, HMI technologies are also challenged to become more human-centric. HMI technologies in DT applications are crucial to support various HMC applications by providing a means to interact with machines and their virtual counterparts. Various technologies are used to visualise data intuitively for users, which can fulfil many roles. As seen in Table 5, these technologies can support operators’ safety and collaboration with machines, while AI algorithms can also be implemented.

One of the most emerging technologies is XR technologies, which encompass augmented reality (AR), virtual reality (VR), and mixed reality (MR). AR is frequently found in the literature, where it is explored for its useful support for human–machine collaborative applications. Thanks to its ability to overlay digital information, it is a useful tool to enhance the safety of workers and increase their productivity and trust of technologies. Augmented reality applications are mostly used in the form of mobile devices, such as phones [83], tablets [55], or glasses.

In contrast, VR provides an immersive virtual environment for users to experience, observe, and interact with virtual objects to perceive the real environment. These virtual models map the sensor data of physical products to reflect their life-cycle process [84]. VR is commonly used in design and simulation use cases, where a fully virtual world is needed to design workshops or test new configurations. In [85], VR was used to generate an industrial human-action-recognition dataset using the DT of an industrial workstation.

One of the common techniques to enable HMI in HMC involves using depth RGB cameras, such as Microsoft Kinect, often used for perceiving the human body and workspace for safety and ergonomics within applications. Using IoT and widespread connectivity, various methods, wireless technologies, and approaches are suggested in scientific articles to offer indoor localisation services, thereby enhancing the services available to users [86]. These localisation technologies include WiFi, Radio Frequency Identification (RFID) devices, Ultra-Wideband (UWB) or Bluetooth Low Energy (BLE).

Natural user interfaces (NUIs) are a type of user interface design that aims to use natural human behaviours and actions for interaction rather than requiring the user to adapt to the technology. NUIs should be designed so that users are able to use them with little to no training [87]. Multimodal interfaces use several ways of HMI, including NUIs, so the users are either free to choose the most convenient method for themselves or use more of them to create better input to process by machines [88]. For example, the authors in [89] used a multimodal interface, which included voice recognition, hand motion recognition, and body posture recognition as the input for deep learning for collaboration scenarios.

**Table 5 sensors-24-02232-t005:** HMI technologies used in DT HMC applications.

HMI Technology		Specific Technique	Task	Application Problem
Touch interfaces		Tablet [55] Phone [83]	Visual augmentation [55,83]	Safety [55,83] HM cooperation [55]
Web interfaces		BLE tags [80]	Indoor positioning [80]	Occupational safety monitoring [80]
Extended reality	VR	HTC Vive [50,56,65] HTC Vive Pro Eye [85] Facebook Oculus [50] Sony PlayStation VR [50] Handheld sensors [65,85]	Training [65] Validation [65] Safe development [56] Data generation [85] Auto-labelling [85] Interaction with virtual environment [74] Robot operation demonstration [50]	Online shopping [74] Human productivity and comfort [50] Human action recognition [85]
	AR	HoloLens 2 [51,79,90] Tablet [55] Phone [83]	Robot teleoperation [51] Visual augmentation [55,83,90] Real-time interaction [79]	Human safety [55,79,83,90] Intuitive human–robot interaction [51] Productivity [79]
	MR	HoloLens 2 [53,75]	Visual augmentation [75] Object manipulation [53]	Human safety [75]
Natural user interfaces	Gestures	HoloLens 2 [53] Kinect [91]	Head gestures [53]	3D object robot manipulation [53,91]
	Motion	Perception Neuron Pro [85] Manus VR Prime II [54] Xsens Awinda [54]	Motion capture [85] Finger tracking [54] Body joint tracking [54]	Human motion recognition [54,85]
	Gaze	Pupil Invisible [54] HoloLens 2 [53]	Object focusing [54] Target tracking [53]	Assembly task precision [54] Interface adaptation [53]
	Voice	HoloLens 2 [53]	MR image capture [53]	3D manipulation [53]

### 4.4. Data Transmission, Storage, and Analysis Technologies

A data-driven digital twin possesses the ability to observe, react, and adjust according to changes in its environment and operational circumstances. Data transmission technologies include wired and wireless transmissions. Both wired and wireless transmissions depend on transmission protocols. Storing collected data for processing, analysis, and management is an essential aspect of database technologies. However, traditional database technologies face challenges with the increasing volume and diversity of data from multiple sources. Therefore, big data has prompted the exploration of alternative solutions, such as distributed file storage (DFS), NoSQL databases, and NewSQL databases. This large volume of data is then preprocessed and analysed for extracting useful information through statistical methods or by database methods, which include multidimensional data analysis and OLAP methods [49]. Nonetheless, AI methods from Section 2.2, such as neural networks or deep learning, can also be used for some analytic tasks. As seen in Table 6, authors often use multiple technologies, and multiple authors do not state specific technologies in their work, but rather, specify the usage of broad technology frameworks such as TCP/IP or cloud databases.

Based on various levels of DTs, edge computing, fog computing, and cloud computing can be used for data transmission, storage, or analysis tasks. Cloud computing provides widespread, easy, and on-demand network access to a pool of resources, shareable as needed, offering high computational and storage capacities at reduced costs. Meanwhile, fog computing extends the cloud’s computing, storage, and networking capabilities to the edge network. Edge computing (EC) then allows data processing to be performed closer to the data sources [92]. As one of the enabling technologies for Industry 5.0 [4,5], EC has already found a number of applications in the literature for network operations, where the edge must be designed efficiently to ensure security, reliability, and privacy.

Since one of the main challenges of DTs is to ensure data flow between the physical and digital counterparts, one of the big research areas that can improve human safety is also task offloading [93]. In many areas, like healthcare, the risk of potential high response latency at the data centre end is critical [94]. Ruggeri et al. [95] proposed a solution that utilises a deep reinforcement learning agent in an HRC scenario, which observes safety and network metrics to decide which model should run on mobile robots and the edge based on network congestion, which greatly improved the safety metrics and reduced the network latency. In the case of task offloading, authors very often focus on Autonomous Mobile Robots (AMRs) because of their limited computation hardware, which poses a challenge in these applications.

Paula Fraga-Lamas et al. [96] proposed a mist/edge computing cyber–physical human-centred system (CPHS) that uses low-cost hardware to detect human proximity to avoid risky situations in industrial scenarios. The proposed system was evaluated in a real-world scenario, where the maximum latency was reduced and low computational complexity was preserved. Research on edge intelligence in DTs that can improve areas such as anomaly detection was also explored [97].

Taking scalability into account, an edge-based twin is most valuable due to its minimal latency, especially when compared to twins based on cloud and edge–cloud configurations [98]. As edge AI and AI-enabled hardware like graphical processing units, such as the Nvidia Jetson series, or AI accelerators, such as Intel Movidius products, continue to evolve, it becomes feasible to break down the DT of an entire factory process into smaller, modular DT processes [44].

Technologies, such as Kubernetes and Docker, streamline container management and workload orchestration, enhancing data handling and processing efficiency in modern computing [99]. The microservices architecture, adopted in the era of cloud computing, facilitates greater customizability, reusability, and scalability by splitting solutions into interconnected applications [100]. This approach, coupled with Kubernetes orchestration and containerisation, significantly boosts deployment efficiency, scalability, flexibility, and reliability in contemporary DT operations and applications [101,102].

Furthermore, incorporating the precise analysis and forecasting strengths of big data, HMC driven by DT technology will become more adaptive and forward-looking, offering significant improvements in various aspects of efficient and accurate management [103].

**Table 6 sensors-24-02232-t006:** Data transmission, storage, and analysis technologies in DT HMC applications.

Category	Technology	Task	Application Problem
Storage	MySQL [55] MongoDB [60] Cloud database [80,104]	CAD, audio, and 3D model files [55] Assembly step executions [60] Robotic arm motion list [104] General system data storage [80]	Human safety [60,80,104] Productivity [60,104]
Data Transmission	TCP/IP [50,55,73,75] Ethernet [55] MQTT [97] Cellular [80] WiFi [55,80] Bluetooth [80]	Physical and digital world communication [50,75] Human–robot Android AR application [55] Robot control movement [73] Occupational safety system [80] Edge intelligence anomaly detection [97]	Human safety [55,75,80] Productivity [55] Human skill level analysis [50] Dynamic changes’ prediction [73] Maintenance [97]
Analysis	Principal component analysis [50] Parameter sensitivity analysis [105]	Dimension reduction [50] Model adaptability enhancement [105]	Human safety [80] Maintenance [105]

## 5. Discussion

The growth of DT applications in HMC and the recent focus on human centricity indicate the potential for human-centric DT applications in various areas of industry. In this review paper, we identified the existing literature and research trends in the HMC domain using the DT as a key enabling technology. We focused on different applications based on enabling technologies identified for the Industry 5.0 era. The limitations of this study may include its main focus on the WoS database and keyword search, which may miss some work on this topic. Based on the literature reviewed, we identified several research gaps.

Firstly, the scarcity of human-centric applications stems from their recent emergence as a focus area, a lack of standardisation, and the complexity of societal and technical challenges. Although creating these human-centric applications lacks standardisation, numerous applications place a significant emphasis on the human aspects. However, most DT applications still focus on productivity as the main goal. In HMC, where humans are part of collaboration processes, focusing on human problems is often inevitable. There is a need to complement DTs with a deep understanding of human behaviours, preferences, and limitations to make the uncertainty of human behaviour less challenging to model. As a result, the following research should start by analysing the impact and significance of the analysed enabling technologies and methods in implementing human-centric DTs across various HMC applications.

Secondly, HMC applications mostly focus on arm manipulators and lack work involving mobile robots. While there are existing industrial applications, research in industrial domains may be proprietary and not widely published due to competitive reasons. At the same time, modelling mobile robots and humans in dynamic environments represents new challenges as opposed to robots whose environment is smaller and does not change much. Therefore, new case studies and experiments are essential to comprehend the practical implications, limitations, and benefits of mobile robots in HMC across various industrial contexts.

Thirdly, the methods for synergistically integrating all enabling technologies in complex systems remain unclear. According to the Industry 5.0 document, all technologies will reveal their full potential when combined with the others to create complex systems. Therefore, a framework should be proposed in future work to integrate the analysed technologies and methods, providing a systematic approach to technology selection and combination for different use cases.

Apart from identified research gaps, the analysed enabling technologies and methods, on their own, have their limitations, which need to be addressed in future research. DTs confront several challenges, including creating accurate virtual models, the absence of a standardised framework for DT development, insufficient training programs, high development costs, and the complexity of accurately modelling human interactions. Furthermore, integrating DT with cyber–physical systems and IoT sensors poses challenges in real-time connectivity and data synchronisation. Therefore, research efforts should prioritise improving the fidelity of DT models, particularly in accurately representing human interactions and physical phenomena. Collaborative efforts between academia, industry, and policymakers will be crucial [44].

Although incorporating HMI technologies into DT applications can enhance user experience and support HMC, achieving true human-centricity in these technologies presents significant challenges. Despite the potential benefits of depth RGB cameras and other IoT devices, it also raises worries regarding privacy and data security when frequently recording sensitive details concerning the movements and interactions of workers, where research can also head in the direction of human-centric privacy- or data-protection systems [106]. Additionally, AR and VR devices have to be designed to be inclusive to all humans and to mitigate any sensory overload or disorientation, particularly in complex industrial environments [107].

Similarly, there is a question on how to solve the biggest challenges to designing, implementing, and deploying fair, ethical, and trustworthy AI. We can address this challenge by focused research questions [108] encompassing issues such as identifying and addressing AI biases, ensuring transparency and explainability, establishing accountability, and developing robust legal and regulatory frameworks.

Moreover, additional limitations and challenges emerge, including reliability and latency issues in data transmission technologies, the scalability constraints of traditional database technologies, and the complexities associated with implementing edge computing, fog computing, and cloud computing solutions. Additionally, complexity challenges persist in managing containerised microservices despite their potential to enhance deployment efficiency.

As many applications achieve better outcomes with a broader array of technologies and data, the use of fusion and multimodal solutions is expected to increase in the coming years. One of the examples is composite AI [109], or composable AI systems [110], which are crucial for the advancement of AI technology, as the modularity and composition of multiple AI models will make creating complex AI systems easier and faster. For instance, in [111], the authors proposed a composite AI model employing a Generative Adversarial Network to predict preemptive migration decisions for proactive fault tolerance in containerised edge deployments.

For advancing video understanding techniques, the authors in [112] developed a Human–Robot Shared Assembly Taxonomy (HR-SAT) for HRC to represent industrial assembly scenarios and human procedural knowledge acquisition, which can be further used for various AI tasks, such as human action recognition and prediction or human–robot knowledge transfer. Federated learning [113] offers an effective method for leveraging the growing processing capabilities of edge devices and vast, varied datasets to develop machine learning models while maintaining data privacy, which could solve some ethics problems and improve human trust in collaboration with machines.

HMC applications involving mobile robots, such as drones or ground AMRs, could also be utilised in the future for their flexibility, manoeuvrability, and adaptability. However, it is harder for applications involving mobile robots to solve safety issues, as their workspace is not constrained to a smaller place than it usually is with assembly cells with cobots. For safety applications involving mobile robots, solutions, such as AR in HMC applications, could be used to visualise the path of mobile robots [83], which will also increase human trust when working in the same workspace and then collaborating on the same tasks. Exploring natural user interfaces (NUIs) for multimodal interaction with mobile robots, such as drones, may also unveil future collaborative applications [114].

Localisation and communication trends such as Visible Light Communication (VLC) or Visible Light Positioning (VLP) [115] are also promising technologies to enable human-centric HMI in future DT applications or where the use of other technologies might be limiting or undesirable.

## 6. Conclusions

This research presents a comprehensive literature review focused on the importance of developing human-centric applications within DT and its technologies for HMC. It examines specific technologies and methods reported in the literature for each technology category. Our focus was on identifying common use cases for DT technologies in the HMC domain, aiming to guide organisations on optimally leveraging these technologies to enhance operations and processes and prioritise human-centric approaches. Despite its importance, the DT, a key component in futuristic synergistic HMC systems, still lacks extensive literature on human-centric applications. This gap partly arises from the absence of standardised frameworks for developing these types of applications. In the coming years, a significant expansion in research on these topics is anticipated, with a focus on addressing the main challenges and exploring enabling technologies and methods. 

## Figures and Tables

**Figure 1 sensors-24-02232-f001:**
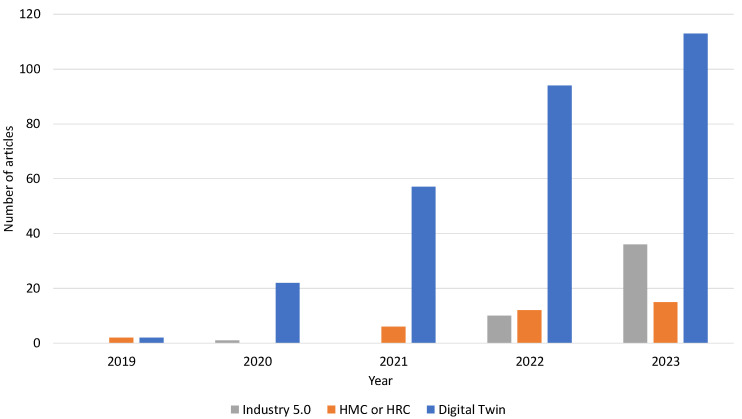
Number of related review and survey articles in the past 5 years.

**Figure 2 sensors-24-02232-f002:**
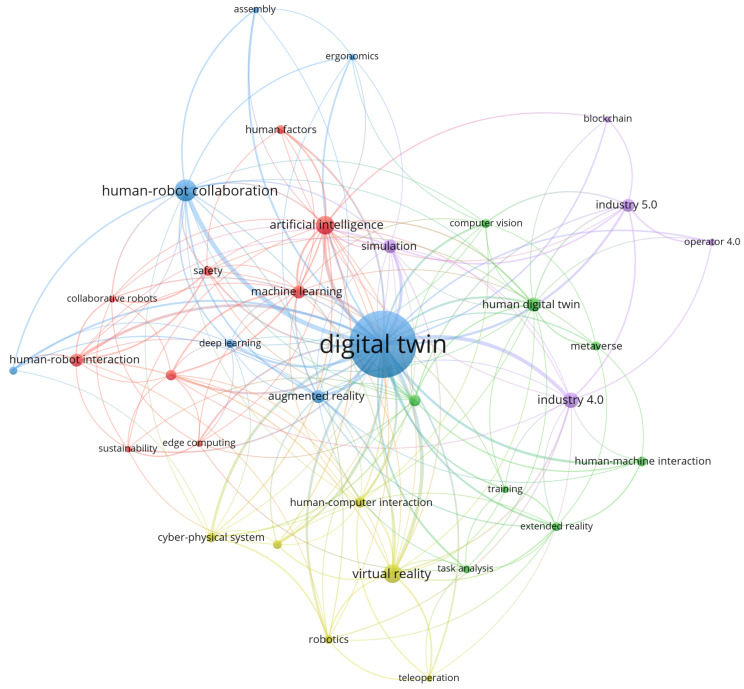
Human-related digital twin keywords co-occurrence network map.

**Table 1 sensors-24-02232-t001:** Comparison of related review and survey papers.

Ref.	Year	Description	ET	EM	UC	HM
[23]	2023	State-of-the-art literature review on human-centric digital twins (HCDTs) and their enabling technologies.	H	H	H	H
[31]	2023	State-of-the-art studies of AR-assisted DTs across different sectors of the industrial field in the design, production, distribution, maintenance, and end-of-life stages.	H	H	H	H
[32]	2023	Recent trends for DT-incorporated robotics.	H	H	H	H
[33]	2023	Literature review on human-centric smart manufacturing to identify promising research topics with high potential for further investigations.	H	H	H	H
[34]	2024	Focus on human centricity as core value of Industry 5.0 and on the concept of human digital twins (HDTs) and their representative applications and technologies	H	H	H	M
[35]	2022	A driver digital twin was introduced to create a more comprehensive model of the human driver.	H	H	H	M
[21]	2023	A systemic review and an in-depth discussion of the key technologies currently being employed in smart manufacturing with HRC systems.	H	M	H	H
[36]	2023	Review on technological aspects of relevant applications dealing with occupational safety and health program issues that can be solved with human-focused DT.	H	M	H	H
[37]	2023	Provides a comprehensive perspective of DTs’ critical design aspects in the broad application areas of human--robot interaction systems.	M	M	M	H
[38]	2022	Research on utilisation of information and communication technologies toward better food sustainability, where humans collaborating with intelligent machines find their place.	M	M	M	M
[39]	2022	Review on simulation platforms and their comparison based on their properties and functionalities from a user’s perspective.	M	H	H	L
[40]	2023	The author examined current DT technology from the viewpoint of human–robot interaction systems.	M	M	M	M
[41]	2022	The integration of human factors into a DT of a city and a human interacting with a DT of objects in the city.	M	M	M	L
[42]	2023	The analysis of the progress of DTs and robotics interfaced with extended reality.	L	L	H	H
[43]	2023	Overview of DT applications within the fields of industry and health. The concept of controlling a rehabilitation exoskeleton via its DT in the VR is presented.	M	M	H	L
[44]	2022	Focus on DT technologies in the manufacturing domain and human–robot collaboration scenarios.	L	L	M	H
[45]	2021	Integration and interaction of human and DT in smart manufacturing systems and current state of the art of DT-based HMI.	L	L	M	L
[46]	2023	The impact of DT technology on industrial manufacturing in the context of Industry 5.0’s potential applications and key modelling technologies is discussed.	H	L	L	L
[47]	2021	Analysis of existing fields of application of DTs for supporting safety management processes and the relation between DTs and safety issues.	L	L	M	L
[48]	2023	Use case review of how human operators affect the performance of cyber–physical systems within a “smart” or “cognitive’” setting.	L	L	L	L

ET—enabling technologies, EM—Enabling Methods, UC—use cases, HM—human–machine focus.

**Table 2 sensors-24-02232-t002:** Co-occurrence network map keywords.

Id	Keyword	Occurrences	Total Link Strength
1	digital twin	229	316
2	human–robot collaboration	58	71
3	virtual reality	36	62
4	artificial intelligence	27	69
5	Industry 4.0	21	40
6	simulation	17	35
7	human digital twin	17	27
8	augmented reality	16	42
9	machine learning	16	36
10	human–robot interaction	16	31
11	Industry 5.0	16	28
12	smart manufacturing	13	38
13	human–computer interaction	11	25
14	Internet of Things	11	25
15	cyber–physical system	10	26
16	safety	10	24
17	human–machine interaction	10	20
18	robotics	9	27
19	human factors	9	19
20	smart city	9	19
21	Metaverse	9	14
22	extended reality	8	21
23	deep learning	8	18
24	computer vision	8	15
25	mixed reality	7	19
26	task analysis	7	17
27	training	6	12
28	Operator 4.0	6	9
29	edge computing	5	17
30	teleoperation	5	17
31	collaborative robotics	5	16
32	sustainability	5	16
33	assembly	5	13
34	ergonomics	5	12
35	blockchain	5	10

**Table 3 sensors-24-02232-t003:** DT tools used in HMC applications.

Digital Twin Tool	Description	Application Areas	Literature Review
Unity [50,51,52,53,54,55,56]	Real-time 3D development platform	Gaming, AR/VR, Automotive	Virtual reality support [50] DT of physical robot [51] DT of virtual space [52] MR system development [53] Human action prediction [54] Safety and productivity [55] Human reaction analysis [56]
Matlab [57,58]	High-level technical computing language	Engineering, Research	Obstacle detection and3D localisation [57] Human digital twin [58]
ROS [55,59,60,61,62]	Middleware for robotics software development	Robotics, automation	Communication [55] Decision making [59] Safety and ergonomics [60] Robot learning [61] Human behaviour [62] Flexible assembly [62]
Gazebo [59,60,61]	Advanced robotics simulation	Robotics, educational research	Decision making [59] Safety and ergonomics [60] Robot learning [61]
Klampt [63]	Versatile motion planning and simulation tool	Robotics, education	Assembly planning [63]
V-REP [64]	Robot dynamics simulator with a rich set of features	Robotics, educational research	Robot control [64]
Siemens NX [65]	Advanced solution for engineering design and simulation	Engineering, manufacturing	Robot programming [65]
Technomatix Process Simulate [22,66,67]	3D simulation of manufacturing processes	Manufacturing, automation	Flexible assembly [66] Design, development, and operation [22,67]

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
