# Peer review of "Towards a Human-Centric Digital Twin for Human–Machine Collaboration: A Review on Enabling Technologies and Methods"

_sensors, 2024, doi:10.3390/s24072232_

Round 1

Reviewer 1 Report

Comments and Suggestions for Authors

While the concept of Digital Twins (DTs) has revolutionized industries and provided invaluable insights into physical entities' behaviour, the focus on productivity and efficiency may overlook the essential aspect of human-centred design. The emphasis on creating digital counterparts primarily for optimizing processes and tasks may inadvertently sideline the human element in Human-Machine Collaboration (HMC). By prioritizing productivity over human-centricity, there's a risk of neglecting the unique capabilities and needs of human workers within manufacturing systems.

Moreover, the current approach to DTs in HMC applications may not adequately address the complexities of human interactions and adaptability in dynamic work environments. While DTs offer a means to simulate and manage complex systems, they must be complemented by a thorough understanding of human behaviors, preferences, and limitations. Without integrating a human-centric approach into the design and implementation of DTs, there's a danger of perpetuating system-centric models that prioritize efficiency at the expense of human well-being and satisfaction.

Therefore, it is imperative to reevaluate the role of DTs in HMC applications and shift towards a more balanced approach that considers both productivity and human-centricity. This necessitates a paradigm shift towards incorporating human-centered design principles into the development of DT-enabled solutions, ensuring that technology serves to enhance human capabilities and experiences rather than overshadowing them. By prioritizing human needs and experiences, we can harness the full potential of DTs to create more inclusive, adaptable, and sustainable manufacturing environments that prioritize both productivity and human well-being.

While it's true that incorporating human-computer interaction (HCI) technologies into Digital Twin (DT) applications can enhance user experience and support Human-Machine Collaboration (HMC), there are significant challenges to achieving true human-centricity in these technologies. Despite the potential benefits of augmented reality (AR), virtual reality (VR), and other immersive technologies in enhancing safety and productivity, there are inherent limitations and risks associated with their implementation.

One of the primary concerns with XR technologies like AR and VR is the potential for sensory overload and disorientation, particularly in complex industrial environments. While these technologies offer immersive experiences, they may also detract from workers' ability to accurately perceive and respond to real-world hazards. Moreover, the reliance on depth RGB cameras and other IoT devices for HMI raises privacy and data security concerns, as these technologies often capture sensitive information about workers' movements and interactions.

Furthermore, the notion of Natural User Interfaces (NUIs) and multimodal interfaces may not always align with the diverse needs and abilities of workers. While these interfaces aim to leverage natural human behaviors for interaction, they may inadvertently exclude individuals with disabilities or those who prefer alternative modes of interaction. Designing interfaces that accommodate a wide range of users' needs and preferences requires careful consideration and inclusive design practices, which may not always be prioritized in DT applications.

Moreover, the integration of AI algorithms into HCI technologies raises ethical considerations regarding algorithmic bias and transparency. Without robust safeguards and accountability mechanisms in place, there's a risk that AI-powered interfaces may perpetuate existing inequalities or inadvertently discriminate against certain groups of workers.

In summary, while HCI technologies hold promise for enhancing HMC and supporting DT applications, it's essential to critically evaluate their impact on workers' safety, privacy, and inclusion. Achieving true human-centricity requires addressing these challenges and prioritizing the well-being and empowerment of workers in the design and implementation of DT-enabled systems.

1. The research gap is not highlighted correctly—research Motivation is lacking.  

2. The research objectives need to be more clear to future readers.

While the growth of Digital Twin (DT) applications in Human-Machine Collaboration (HMC) and the recent emphasis on human-centricity are indeed promising, there are several limitations and challenges that need to be addressed to realize the full potential of human-centric DT applications.

Firstly, despite the increasing focus on human-centric applications, there remains a significant gap between theory and practice. While there is recognition of the importance of prioritizing human needs in DT applications, the practical implementation of human-centric design principles remains a challenge. Many existing DT applications still prioritize productivity and efficiency over human well-being, highlighting a disconnect between theoretical ideals and practical implementation.

Secondly, the focus on arm manipulators in HMC applications neglects the potential of mobile robots, such as drones or ground Autonomous Mobile Robots (AMRs), which offer greater flexibility and adaptability in dynamic environments. However, integrating mobile robots into collaborative workflows presents unique challenges, particularly in terms of safety and coordination. Without robust frameworks and guidelines for incorporating mobile robots into HMC applications, their full potential may not be realized.

Thirdly, achieving synergistic usage of enabling technologies in DT applications remains a significant challenge. While there is acknowledgment of the importance of combining multiple technologies to create complex systems, there is a lack of clear guidance on which technologies to combine and how to effectively integrate them. As a result, many DT applications may not fully leverage the capabilities of available technologies, limiting their effectiveness and potential impact.

Moreover, while fusion and multimodal solutions hold promise for advancing DT applications, there are significant technical and practical hurdles that need to be overcome. For example, composite AI systems and federated learning offer exciting opportunities for enhancing AI capabilities, but they also raise concerns about data privacy and ethical implications. Without addressing these challenges, the widespread adoption of fusion and multimodal solutions may be hindered.

In summary, while the growth of human-centric DT applications in HMC is a positive development, it is essential to address the aforementioned limitations and challenges to realize their full potential. By bridging the gap between theory and practice, incorporating mobile robots effectively, and leveraging enabling technologies synergistically, human-centric DT applications can truly revolutionize industries and enhance human-machine collaboration.

While the literature review conducted in this research provides valuable insights into the importance of developing human-centric applications within Digital Twin (DT) technologies for Human-Machine Collaboration (HMC), there are several limitations and challenges that need to be addressed to advance this field effectively.

Firstly, while the review identifies common use cases for DT technologies in the HMC domain, there remains a lack of comprehensive literature on human-centric applications. Despite the growing recognition of the importance of prioritizing human needs in DT applications, there is still a scarcity of research that specifically focuses on the development of human-centric DT solutions. Without a robust body of literature in this area, organizations may struggle to implement human-centric DT applications effectively.

Secondly, the lack of standardized frameworks for creating human-centric DT applications presents a significant barrier to progress in this field. Without clear guidelines and methodologies for developing human-centric DT solutions, organizations may face challenges in designing, implementing, and evaluating these applications. Standardized frameworks are essential for ensuring consistency, interoperability, and scalability across different human-centric DT projects.

Moreover, while the research anticipates considerable expansion in this area in the coming years, there is a need for more focused efforts to address the main challenges and identify enabling technologies and methods. Merely expecting a broad expansion of research may not be sufficient to overcome the existing barriers and propel the field of human-centric DT applications forward.

In summary, while the literature review highlights the importance of developing human-centric applications within DT technologies for HMC, there are critical gaps that need to be addressed to advance this field effectively. By addressing the lack of comprehensive literature, establishing standardized frameworks, and focusing research efforts on key challenges and enabling technologies, the field can move closer to realizing the full potential of human-centric DT applications in enhancing human-machine collaboration.

Comments on the Quality of English Language

Minor editing is required. 

Author Response

Dear reviewer, we would like to thank You for the insightful comments. All comments were considered carefully and the revisions are summarised as follows. We are happy to respond to any further comments both the editor and the reviewers may raise hereafter.

Reviewer 2 Report

Comments and Suggestions for Authors

In this paper, a review on enabling technologies and methods is provided for human-centric digital twin and human-machine collaboration. However, there are some problems which must be solved before it is considered for publication. The detailed comments are given as follows:

1. In the introduction section there should be a brief description of the background and importance of the human-centric digital twin and human-machine collaboration.

2. In the second section, the author should give a strong link between Industry 5.0, human-machine collaboration and digital twin, rather than just presenting each part separately.

3. In the fourth section, digital twin simulation and human-computer interaction are part of artificial intelligence. In addition, data transfer, storage and analysis techniques are also included in digital twin simulation, artificial intelligence and human-computer interaction. The authors should clarify these concepts.

4. It is suggested that the authors analyze the limitations of digital twins and human-computer interaction in past research and suggest recommendations and directions for future research.

5. The authors should carefully check this paper for grammatical and writing errors.

Comments on the Quality of English Language

None

Author Response

(The authors gave the same response as above.)

Reviewer 3 Report

Comments and Suggestions for Authors

A good review paper and the reference materials presented are up-to-date. It will be useful to the researchers in this field. 

Can consider adding topics such as Kubernetes (K8S), Containerization and Microservices in DT operations. 

Comments on the Quality of English Language

There are minor grammatical errors which need to be corrected. I did not list them out here, but authors are advised to read through the manuscript once again to identify their whereabouts. 

Author Response

(The authors gave the same response as above.)

Round 2

Reviewer 1 Report

Comments and Suggestions for Authors

I am happy to see the revisions for the manuscript.

Comments on the Quality of English Language

Minor English editing is required.